# The molecular basis of pyrazinamide activity on *Mycobacterium tuberculosis PanD*

Qingan Sun [1], Xiaojun Li[1], Lisa M. Perez [2], Wanliang Shi[3], Ying Zhang[3] & James C. Sacchettini[1]*

Pyrazinamide has been a mainstay in the multidrug regimens used to treat tuberculosis. It is active against the persistent, non-replicating mycobacteria responsible for the protracted therapy required to cure tuberculosis. Pyrazinamide is a pro-drug that is converted into pyrazinoic acid (POA) by pyrazinamidase, however, the exact target of the drug has been difficult to determine. Here we show the enzyme PanD binds POA in its active site in a manner consistent with competitive inhibition. The active site is not directly accessible to the inhibitor, suggesting the protein must undergo a conformational change to bind the inhibitor. This is consistent with the slow binding kinetics we determined for POA. Drug-resistant mutations cluster near loops that lay on top of the active site. These resistant mutants show reduced affinity and residence time of POA consistent with a model where resistance occurs by destabilizing the closed conformation of the active site.

[1] Department of Biochemistry and Biophysics, Texas A&M University, College Station, TX, USA. [2] Laboratory for Molecular Simulation, Texas A&M University, College Station, TX, USA. [3] Department of Molecular Microbiology and Immunology, Bloomberg School of Public Health, Johns Hopkins University, Baltimore, MD 21205, USA. *email: sacchett@tamu.edu

In 2018, tuberculosis (TB) claimed more lives than any other infectious disease[1]. This is primarily because TB is rampant precisely in the areas where it is most challenging to treat due to the poverty of those areas and the lengthy treatment course. Adding to the difficulty in effective treatment is widespread drug resistance. Of the four front-line anti-TB drugs (isoniazid, rifampin, pyrazinamide (PZA), and ethambutol), PZA has been an irreplaceable component in clinical TB regimens due to its unique activity against persisters in chronic *Mtb* infections and ability to shorten treatment times[2]. It is for these reasons that PZA is recommended by the WHO as an integral component in the treatment of multidrug resistant TB[3].

Despite the clinical importance of PZA and the early recognition of its anti-TB activity back in the 1950s, its mechanism of action has not been fully understood[2,4]. This is primarily because in vitro mode-of-action experiments are complicated by the fact that PZA is not active against *Mtb* cultures grown in normal media[5]. It is active under mildly acidic pH, but even then it is not very potent with minimum inhibitory concentrations between 0.4–1.6 mM[6,7]. What is certain is that PZA is a prodrug that is converted into pyrazinoic acid (POA) by the enzyme pyrazinamidase, PncA (Rv2043c)[8], and that most of the PZA clinical resistance arises from loss of function mutations in *pncA*. However, multiple mechanisms of action and targets have been reported for PZA[9–18]. Recently, Zhang et al. reported several PZA resistant mutations that mapped to the *Mtb panD* gene[19,20], suggesting it was a target of PZA. PanD is an aspartate decarboxylase responsible for the formation of β-alanine from L-aspartate, which is part of the pantothenate biosynthetic pathway, essential for vitamin B5 and coenzyme A biosynthesis in *Mtb*[21].

Here, we present biochemical and structural evidence that POA inhibits *Mtb* PanD in a competitive fashion and that it binds with a high degree of complementarity to the active site of the enzyme. The binding affinity is consistent with the potency of PZA against *Mtb*. In addition, the majority of the PZA drug-resistant mutations are located close to the catalytic center interacting with amino acids on two loops that enclose the active site. These results serve to solidify the prior clinical sequencing data that implicated PanD as the target for PZA.

## Results

**POA is a competitive inhibitor of PanD**. Using purified recombinant PanD, we have characterized the inhibition of the enzyme by POA with a coupled enzyme assay. The data clearly showed that POA inhibition is competitive with aspartate (Fig. 1a) in contrast to previously published results, placing it distant from the active site[22]. POA has a $K_i$ of 0.78 (0.05) mM, which is comparable to the $K_M$ of 1.08 (0.06) mM for the substrate aspartate. We also tested PZA, nicotinic acid (NA), and two POA analogs with chlorine atom at the 5-(para) or 6-(meta) position (Fig. 1b). None of these compounds inhibited *Mtb* PanD at concentrations up to 20 mM, except 6-Cl-POA which was also competitive and slightly less potent compared to POA ($K_i = 1.00$ (0.04) mM, Fig. 1c).

We next used isothermal titration calorimetry (ITC) to directly measure POA's binding affinity. POA, as a weak acid, can generate robust ITC signals on its own due to pH-dependent proton dissociation[12]. We also observed that *Mtb* PanD is sensitive to acidic pH, and it begins to precipitate <pH 7.0. To prevent pH related artifacts in the ITC, we buffered both the POA and the protein in 100 mM Tris buffer at pH 7.5 (Fig. 1d and Supplementary Fig. 1). The ITC data fit to a single-site model, with a $K_d$ for POA of 0.71 (0.03) mM ($n = 5$) and that of 6-Cl-POA as 1.09 (0.04) mM ($n = 3$). These values are in very good agreement with the $K_i$ values of 0.78 mM and 1.0 mM described

above. We saw no binding of NA, indicating that the single-atom change on the pyrazine ring (N1- > C) abolished the interaction between POA and *Mtb* PanD. The ITC results for both POA and 6-Cl-POA, showed that the measured $\Delta H$ was negative for both inhibitors (−4200 (200) cal mol$^{-1}$ and −3300 (100) cal mol$^{-1}$), while the $\Delta S$ was relatively small (0.1 (0.7) cal mol$^{-1}$ deg$^{-1}$ and 2.2 (0.5) cal mol$^{-1}$ deg$^{-1}$). This indicated that the binding was enthalpy driven and that the POA and 6-Cl-POA interactions with the enzyme were primarily electrostatic or hydrogen bonds. This is in fact what we observed in the structure: hydrogen bonds and electrostatic interactions are critical for binding of POA to PanD, as described below. It is difficult to interpret the ITC results from a previous publication where POA was reported to have a $K_d$ of 6.1 μM for *Mtb* PanD, which was based on ITC in unbuffered water[22]. The pH of the POA in unbuffered water is <3.0 which brings into question the physiological relevance of the result.

To confirm our ITC binding data and to determine the on-rate and off-rate of binding, we employed an orthogonal method, BioLayer interferometry (BLI)[23], to measure POA binding to PanD. We monitored the real-time association and dissociation between POA, and *Mtb* PanD on the surface of the biosensor. The $k_{on}$ rate for POA is 3.5 (0.6) M$^{-1}$ s$^{-1}$, and the $k_{off}$ is 0.0027 (0.0001) s$^{-1}$. The $K_d$ was derived from $k_{off}/k_{on}$ as 0.8 (0.1) mM and matched well with the affinity we measured through ITC (Supplementary Fig. 2). The relatively high $K_d$ is primarily related to the slow on-rate of POA. In short, POA binds *Mtb* PanD with a slow on-rate, but POA–PanD complex is very tight once formed, as evidenced by the very slow off-rate. The observed kinetics of the POA–PanD interaction may help to explain the paradox of PZA as a slow-acting, low-potency, but highly effective TB drug.

**POA binds to the active site of PanD**. Crystals of *Mtb* PanD were produced using the hanging-drop vapor diffusion method. The crystal structures of apo *Mtb* PanD and the *Mtb* PanD: POA complex were solved using molecular replacement and both structures were refined to 2.7 Å resolution (Supplementary Note 1). The statistics describing the quality of the diffraction data and structures are in Supplementary Table 1. The first 115 residues were clearly visible in the electron density map, while the last 24 residues could not be built due to the flexibility at the C-terminus. PanD is a proenzyme and in the structure, we observed the processed active enzyme after cleavage between Gly24 and Ser25, and modification of Ser25 to a pyruvoyl group (Pyr25). The active protein consists of a β chain (Met1-Gly24) and an α chain (Pyr25 to ILE115). In the POA complex structure, there was one molecule of POA bound to each *Mtb* PanD active site, which was located at the interface between two adjacent subunits of the PanD tetramer (Fig. 2a). The density for POA was clearly visible in a difference electron density map made by subtracting the structure factor amplitudes of the apo crystal diffraction data from that of the PanD:POA complex, as well as a polder OMIT map[24] (Fig. 2b, c). The superimposition of the PanD:POA structure with the crystal structure of *Helicobacter pylori* PanD bound to a substrate analog, isoasparagine (pdb: 1UHE[25]), showed that there was good agreement in the hydrogen bonding of the inhibitor and substrate analog to the protein. Specifically, an oxygen from the carboxylate of POA is positioned very close to an oxygen of the isoasparagine γ-carboxyl group and forms hydrogen bonding interactions with the guanidinium group of Arg54* (Supplementary Fig. 3)[25].

Because of the internal symmetry of the POA there are two ways to fit POA to the electron density. We choose to position the molecule based on optimal hydrogen bonding (Fig. 2b).

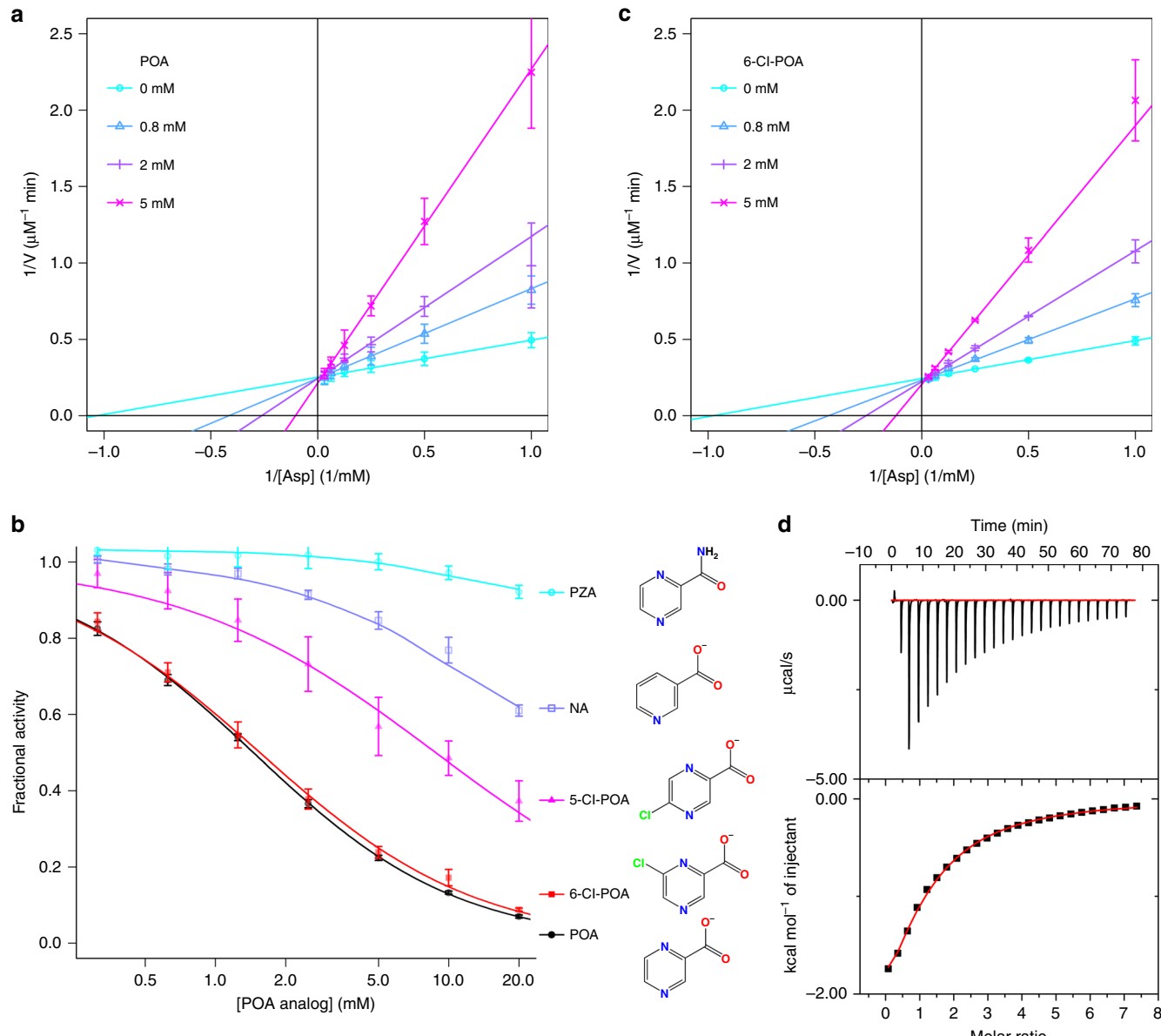

**Fig. 1 Biochemical characterization of the interaction between *Mtb* PanD and POA. a** POA showed competitive inhibition of *Mtb* PanD. Lineweaver–Burk plots of *Mtb* PanD activity in the presence of various concentrations of POA prepared in triplicate. The data were fitted with a competitive inhibition model, yielding $K_i = 0.78$ (0.05) mM, $K_M = 1.08$ (0.06) mM, and $k_{cat} = 0.330$ (0.006) s$^{-1}$. **b** Inhibition of PanD activity with POA analogs. Dose–response curves were measured using the described PanD assay with pyrazinamide (PZA), nicotinic acid (NA), 5-Cl-POA, 6-Cl-POA, and POA. There were six replicates. **c** 6-Cl-POA inhibits *Mtb* PanD comparatively to POA. The plot was indicative of a competitive model of inhibition with $K_i = 1.00$ (0.04) mM, $K_M = 1.12$ (0.04 mM), and $k_{cat} = 0.350$ (0.003) s$^{-1}$. **d** Isotherm calorimetry of *Mtb* PanD with POA. The top panel shows the heat released per injection of inhibitor, as μcal s$^{-1}$; while the bottom panel shows the change in enthalpy (kcal mole$^{-1}$) as a function of the molar ratio of POA to PanD. Titrations were performed at 20 °C using 100 mM Tris buffer (pH 7.5) for both the protein solution and POA titrant. The data were fitted with a single-site binding model. From five separate experiments, it was calculated that $K_d = 0.71$ (0.03) mM, $\Delta H = -4200$ (200) cal mol$^{-1}$, $\Delta S = 0.1$ (0.7) cal mol$^{-1}$ deg$^{-1}$. Error bars were defined as standard deviations. Source data are provided as a Source Data file.

This fit was confirmed when we docked POA into this binding pocket with Glide[26]. The prediction put POA in the same pose as in the crystal structure with all the interactions preserved, the root-mean-square difference on non-hydrogen atoms of POA was only 0.29 Å. The molecular surface of the active site indicated there was room to add a substituent to the pyrazine ring of POA at the 6-position based on the pose we selected. This was consistent with the finding that 6-Cl POA was an inhibitor of PanD and 5-Cl POA showed no inhibition. We cocrystallized *Mtb* PanD with the 6-Cl POA analog and refined the structure to 2.25 Å (Supplementary Table 1).

A polder OMIT map showed clear electron density for the 6-Cl and it was oriented with the pyrazine ring in the same orientation as we fit POA (Fig. 2d, e).

The POA binding pocket is relatively small (206 Å$^3$) and not easily accessible, as there is only a narrow tunnel leading from the active site to the inside barrel of the PanD tetramer. With a diameter of ~3.4 Å, neither the substrate nor inhibitor could pass through the tunnel. The chlorine atom of 6-Cl-POA resides in this tunnel. There is also a small protrusion of space on the opposite side of the tunnel that extends toward the outside surface of the tetramer, but it is blocked from being open to

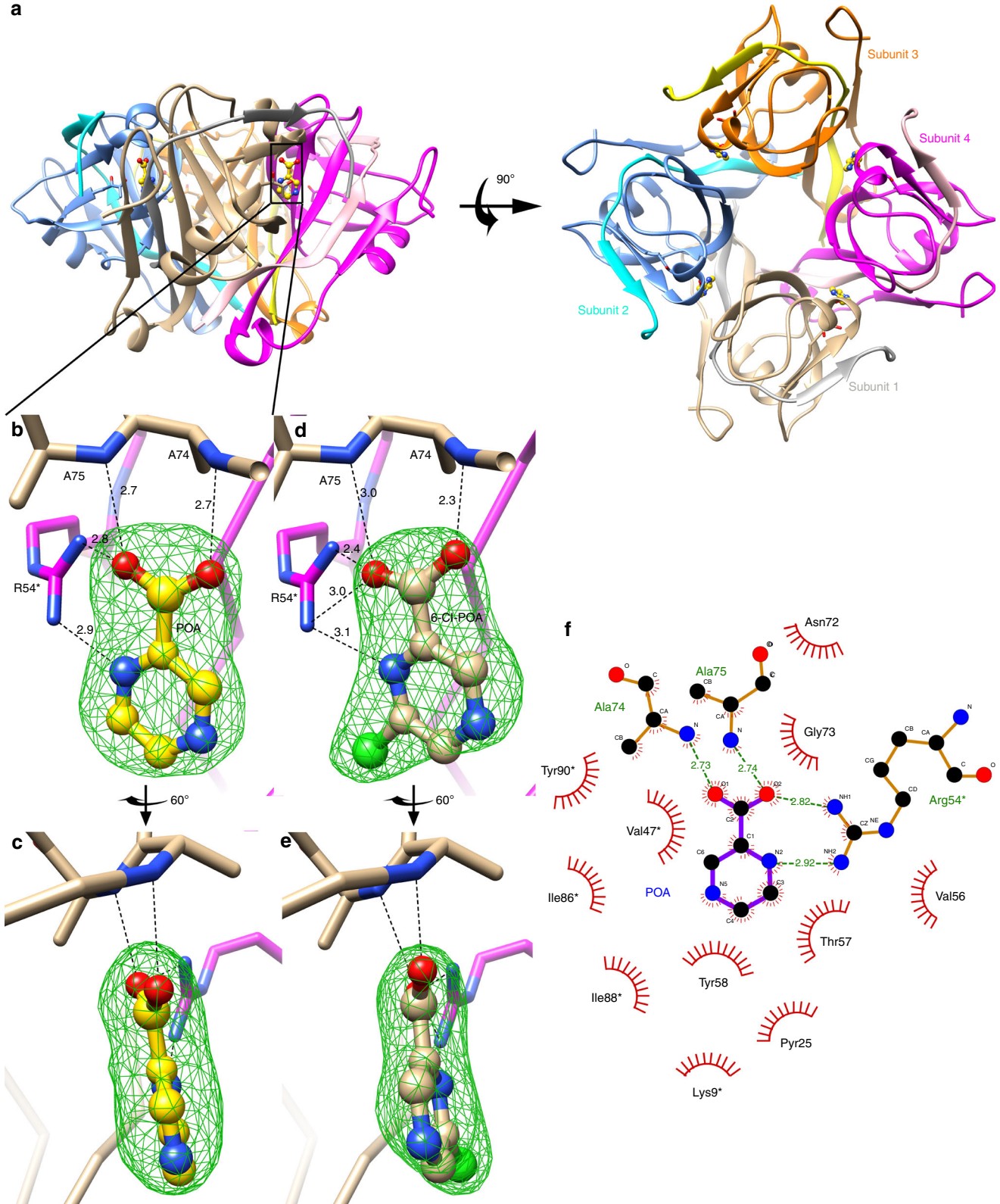

**Fig. 2 *Mtb* PanD–POA interaction in the X-ray crystal structure. a** POA binds *Mtb* PanD tetramer. The four subunits are shown in tan/silver, blue/cyan, orange/yellow, and magenta/pink. POA, shown in ball-and-stick, is bound to the active site at the interface of two PanD subunits. **b** Close up of the POA binding site. POA is shown with ball-and-stick with carbon in gold. The green mesh shows the polder omit map around POA at 3.5 σ. Hydrogen bonds around POA are shown in dash lines. **c** The view is turned 60° vertically. **d**, **e** 6-Cl-POA polder omit electron density. The PanD:6-Cl-POA is shown in the same view as PanD:POA in Fig. 2b, c. **f** POA shows extensive hydrogen bonding interactions with Mtb PanD. The interactions were analyzed and rendered with LigPlot[+ 27].

the surface by the two C-terminal loops of the α and β chain. While we cannot be certain through which portal the substrate or inhibitor enters the active site, the protein must undergo a significant conformational change to allow substrate or POA access to the active site. This conformational change requirement for binding of molecules to the active site may account for the slow on- and off-rate of POA binding.

There were four hydrogen bonds that contribute significantly to the binding between POA and PanD: two between POAs carboxylate oxygens and the amide groups on the backbone of Ala74 (2.7 Å) and Ala75 (2.7 Å), one between a carboxylate oxygen to the guanidinium group of Arg54* (2.8 Å), and another between the N1 on the POA pyrazine ring and Arg54* (2.9 Å, Fig. 2f). When we mutated[27] Arg54 to alanine, the inactive enzyme showed no binding to POA in the ITC experiment. This H-bond explains the selectivity for the pyrazine ring and carboxylate consistent with the lack of inhibitory activity of both NA and PZA. Thus, POA binds to the active site of $Mtb$ PanD interacting with the same groups as the substrate, and this is consistent with a competitive mode of inhibition.

### PZA/POA resistance arises from mutations on active site loops.
To investigate the mechanism of resistance, we purified recombinant protein from two of the most well-studied resistant mutants, H21R and M117I[11,19,20]. Using ITC, we found that the $K_d$ for POA binding to H21R was 2.84 (0.06) mM (Fig. 3a), and the $K_i$ for POA inhibition of H21R was 3.0 (0.1) mM (Fig. 3b, c). The $K_d$ for the mutant was ~2.1 mM higher than for the wild-type enzyme. However, resistance to POA came with a cost of enzyme activity: its $k_{cat}$ was 0.184 (0.003) s$^{-1}$ compared to 0.330 (0.006) s$^{-1}$ for the wild type, $K_M$ was 5.5 (0.2) mM compared to 1.08 (0.06) mM for the wild type. We found that POA also bound to the M117I protein with significantly lower affinity compared to the wild type. The $K_d$ was ~200 μM higher for the mutant than the wild type ($K_d$ = 0.90 (0.09) mM, $n$ = 4; Fig. 3a). The kinetics analysis showed that the POA off-rate measured by BLI ($k_{off}$ = 0.051 (0.008) s$^{-1}$ and 0.06 (0.01) s$^{-1}$ for the H21R and M117I mutants, respectively) was much faster than the wild-type 0.0027 (0.0001) s$^{-1}$ (Fig. 3d). This suggests that the resistance of these two mutations is dependent on a combination of lower affinity as related to the decreased residence time of POA on PanD.

We crystallized and solved the structure of the M117I mutant protein. PanD M117I crystallized in the I422 space group and diffracted to 2.33 Å resolution (Supplementary Table 1). The structure was very similar to the wild-type protein structure with a root-mean-square difference in the cα positions of 0.32 Å. The crystallization condition contained tartrate, a known inhibitor of the enzyme[28]. A molecule of tartrate fit well into electron density in the active site that could not be accounted for by the protein. The resulting structure showed reasonable interactions between tartrate and active site amino acids. In the mutant structure, we could fit an additional 11 residues at C-terminus that were not visible in electron density map for the wild-type protein. The ordered C-terminus enabled us to map several additional resistant mutation sites onto the structure, including M117, that were not visible in the wild-type structure (Fig. 3e). While none of the mutations were in amino acids that directly contacted POA, most were either on the C-terminal loops of the α and β chain or directly interacting with residues from these loops. As described above, these loops appear to form a flexible lid over the active site that sequesters the substrate from the solvent. That they are not directly in the active site is expected given the relatively small volume of the active site, where any mutation in the active site

would likely be deleterious to the normal catalytic activity of the enzyme.

The biochemical and structural data serve to establish that PanD is the primary target of PZA. Our results show that resistance to PZA in PanD mutants primarily comes from alterations in two loops covering the active site that have subtle, but significant effects on the affinity and residence time of POA on the enzyme rather than from directly affecting POA binding interactions.

## Methods

**Protein expression and purification**. Wild-type $Mtb$ $panD$ gene ($Rv3601c$) was PCR-amplified from $Mtb$ genomic DNA and cloned into the pET21c(+)vector[29]. All mutations were introduced through primers (Supplementary Table 2) using the DNA assembly kit (New England Biolabs). The plasmids were transformed into $Escherichia$ $coli$ BL21(DE3) cells for protein production. Protein expression was induced with 0.5 mM IPTG at OD$_{600}$ 0.6 ~ 1, then the cells were incubated at 18 °C overnight before harvesting by centrifugation. The cell pellets were resuspended in Ni-Buffer A (20 mM Tris, pH 8, 150 mM NaCl, 1 mM 2-mer-captoethanol, and 5% glycerol) with an additional 1 mM PMSF and 1 mM benzamidine, then mechanically lysed using a microfluidics with pressure setting at 20,000 psi. The lysate was cleared by centrifugation at 39,000 × $g$ for 1 h before application to HisTrap FF column (GE Healthcare). After extensive washing, the protein was eluted with linear gradient to 100% Ni-Buffer B (Ni-Buffer A + 500 mM imidazole). The fractions with PanD were pooled and dialyzed against 2 L Q-Buffer A (10 mM Tris, pH 8, 50 mM NaCl, 2 mM DTT, and 5% glycerol) before application to POROS HQ column (Thermofisher). Size-exclusion chromatography was carried out with a Sephacryl s-300 column (GE Healthcare) in buffer containing 20 mM Tris, pH 8, 50 mM KCl, and 0.5 mM TCEP. The final protein was concentrated to ~100 mg ml$^{-1}$, flash-frozen with liquid nitrogen, and kept at −80 °C freezer.

**BLI**. POA binding kinetics on $Mtb$ PanD were characterized using an Octet RED96 System (fortéBIO, Pall Corp., USA). All reagents were buffered with 100 mM Tris, pH 7.5. Biotinylated PanD were loaded onto eight High Precision Streptavidin (SAX) Biosensors (18–0037, fortéBIO, Pall Corp., USA), followed by a blocking step with biocytin. Then, a second baseline step was performed before samples (POA or PZA) at various concentrations were associated. The association and dissociation profiles were measured. Another set of eight biosensors were loaded with biocytin, processed in the same manner and served as the references (Supplementary Fig. 2a). Experiments were performed using the kinetics mode, at 28 °C and sample plates were agitated at 1000 rpm. The data were analyzed with python/R script with the following routine: subtract eight references curves from the corresponding sample curves. Then the resulting curve with 0 mM POA serving as the baseline was subtracted from the curves with POA ranging from 16 mM to 0.25 mM. The baseline-corrected data were normalized with the second baseline step. The final association and dissociation curves were fitted with 'nls' function in R using Eq. (1):

$$\text{Response}(x_i, t) = \begin{cases} R_a(x_i, t) + a_i + b_i \times t, \ 0 < t < T_a \\ R_d(x_i, t - T_a) + c_i + d_i \times (t - T_a), \ t > T_a \end{cases} \quad (1)$$

where

$$R_a(x_i, t) = \frac{R_{Max} \times \left(1 - e^{-(k_{on} \times x_i + k_{off}) \times t}\right)}{1 + \frac{k_{off}}{k_{on} \times x_i}}, \quad (2)$$

$$R_d(x_i, t) = R_a(x_i, T_a) \times e^{-k_{off} \times (t - T_a)} \quad (3)$$

Here, $x_i$ is POA concentration; $T_a$ is the association period (600 s).

The R codes were built into an R package that we named 'smoke', and deposited in the CRAN repository [https://CRAN.R-project.org/package=smoke].

**Enzymatic assays**. The aspartate decarboxylase activity of PanD was coupled to the pantoate-β-alanine ligase (PanC) reaction. The generation of pyrophosphate from this reaction was detected with the EnzChek$^{TM}$ Pyrophosphate Assay Kit (ThermoFisher E6645). The reaction volume was 100 μl. In each reaction, there is 0.2 μM $Mtb$ PanD, 3 μM $Mtb$ PanC, 2 mM ATP, 2 mM pantoate, 100 mM HEPES (pH7), and 10 mM MgCl$_2$. All reactions were carried out at 28 °C in a CLAIROstar plate reader (BMG Labtech) and the absorbance at 360 nm was measured. The data analysis and model fitting were done with R scripting [http://www.R-project.org].

**ITC**. ITC experiments were performed using a MicroCal iTC200 (GE). Both protein and ligands were in a buffer of 100 mM Tris, pH 7.5. The protein concentration in the cell was 0.5 mM, and the ligand concentration was 20 mM. The titration was performed at 20 °C. The data were analyzed with the Origin 7.0 software and fitted with a single-site binding model.

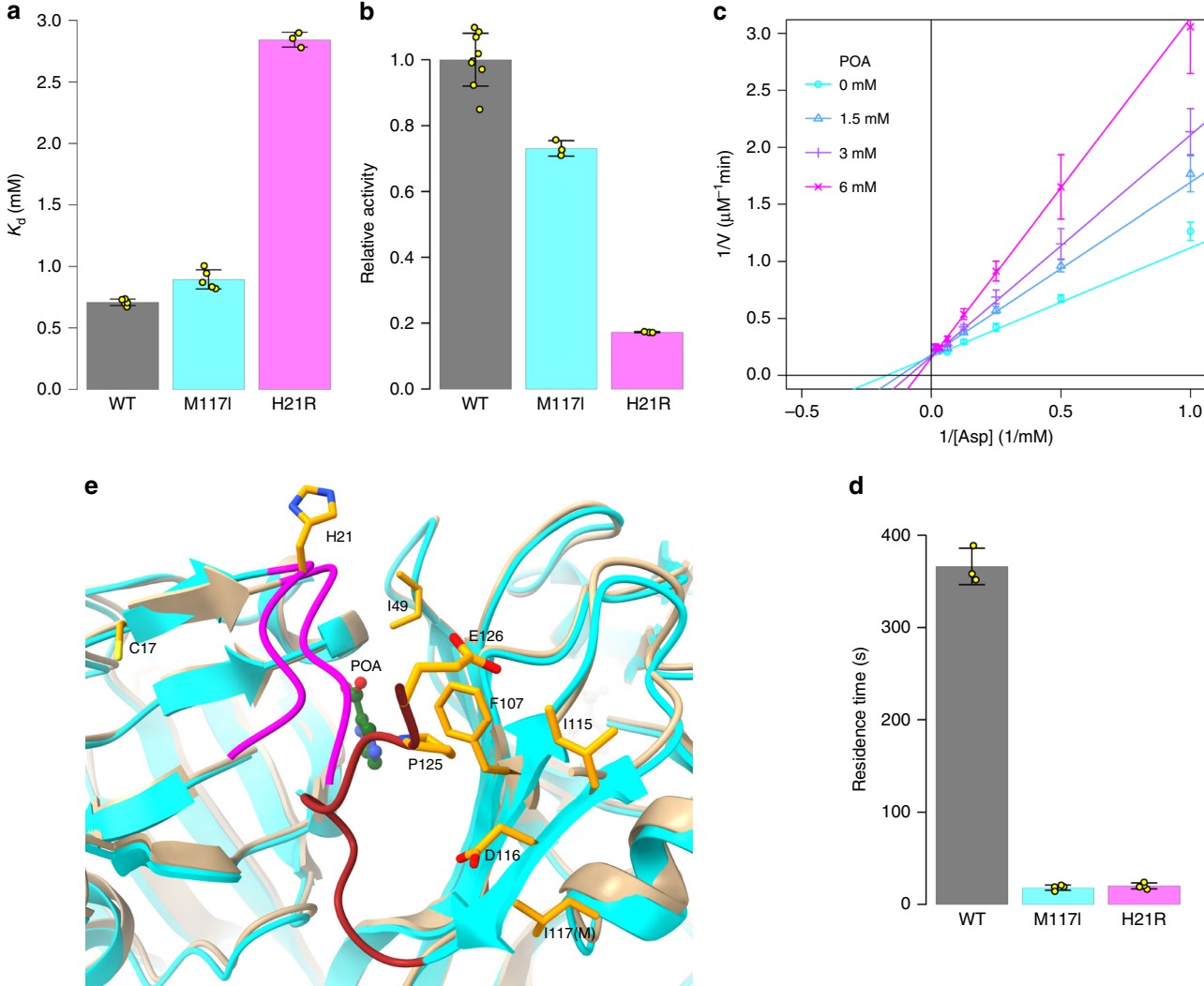

**Fig. 3 POA$^r$ mutations on *Mtb* PanD. a** H21R showed resistance to POA. Lineweaver–Burk plots of H21R activity in the presence of various concentrations of POA prepared separately in quadruplicate. The data were fitted with a competitive inhibition model, yielding $K_i = 3.0$ (0.1) mM, $K_M = 5.5$ (0.2) mM, and $k_{cat} = 0.184$ (0.003) $s^{-1}$. **b** The relative enzyme activity of *Mtb* PanD proteins. The aspartate concentration was 1 mM in the assay. **c** The affinity of *Mtb* PanD:POA was measured with ITC. POA at 40 mM was titrated into 0.5 mM H21R PanD, while 20 mM POA was used in the wild type and M117I experiments. **d** Residence time of POA–panD measured with BLI. **e** Mapping of POA$^r$ sites on M117I and wild-type PanD:POA structures. A ribbon structure of the M117I mutant colored in cyan is superimposed on to the wild-type PanD ribbon in tan. Side chains of the reported resistant mutants mapped to the M117I structure are shown in orange. POA is shown in green. The β-chain C-terminal loop Leu20-Gly24 is highlighted in magenta, while the α-chain loop His119-Glu126 is in brown. Error bars were defined as standard deviations. Source data are provided as a Source Data file.

**Crystallization and data collection**. Protein stocks stored in −80 °C were quickly thawed and diluted to 20 mg ml$^{-1}$ for all crystallization trials. Hanging-drop vapor method was used at 16 °C to produce PanD crystals. For 6-Cl-POA complex crystals, 20 mM ligand was premixed in the protein solution. Wild-type *Mtb* PanD crystals were produced in mother liquor containing 15–18% PEG 3350, 200 mM ammonium chloride, and 100 mM MES/HEPES pH 6.5–7. The cryo-protectant containing 20% PEG 3350, 200 mM ammonium chloride, 100 mM HEPES pH 7, and 15% glycerol. A total of 10 mM POA or 6-Cl-POA was mixed in the cryo-protectant for the respective complex crystals. The crystals were immediately flash cooled in liquid nitrogen. The diffraction data were collected at APS 23ID and processed with Proteum 3 software (Bruker). PanD M117I crystallized at 20 mg ml$^{-1}$ in mother liquor containing 100 mM ammonium tartrate and 12% PEG3350. Additional 15% glycerol was added into the mother liquor as a cryo-protectant to flash cool crystals. The diffraction data were collected at APS 19ID and processed with HKL3000[30].

**Structural determination and refinement**. The initial Apo *Mtb* PanD model was from the molecular replacement with pdb ID: 2C45 as the search template in phaser[31]. This model was used as the template in molecular replacement for all the other PanD structures in this study. Structure refinement was performed in Refmac5 and Phenix refine, combined with manual correction and rebuild

iteratively. The final structural model of PanD:POA complex was refined to Rwork/Rfree of 0.1992/0.2478. In the final Ramachandran plot, 99.55% and 0.45% of residues were in the favored and allowed regions, respectively. The data collection and refinement statistics are listed in Supplementary Table 1. All structure figures were drawn in Chimera.

**Reporting summary**. Further information on research design is available in the Nature Research Reporting Summary linked to this article.

## Data availability
Coordinates and structure factors have been deposited in the Protein Data Bank under accession codes: 6OZ8 for Apo_PanD, 6OYY for POA:PanD, 6P02 for 6-Cl-POA: PanD, and 6P1Y for M117I mutant. Figure data can be found at [https://github.com/quinsun/mtbPanD]. Other data are available from the corresponding author upon request.

## Code availability
Code used for analyzing BLI kinetics and the related dataset has been built into an R package, we named 'smoke', and deposited at the CRAN repository [https://CRAN.R-project.org/package=smoke].

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

## Acknowledgements

Funding for this work came from the TB Structural Genomics Consortium, P01AI095208 from NIAID, NIH (J.C.S.), grant A-0015 from the Welch Foundation (J.C.S.), and a grant from the Chancellor Research Initiative-Texas A&M System (J.C.S.). Results shown in this report are derived from work performed at Argonne National Laboratory, a U.S. Department of Energy (DOE) Office of Science User Facility operated for the DOE Office of Science by Argonne National Laboratory. Data were collected from the Structural Biology Center (SBC) at the Advanced Photon Source (Argonne National Laboratory). SBC-CAT is operated by U. Chicago Argonne, LLC, for the U.S. Department of Energy, Office of Biological and Environmental Research under contract DE-AC02-06CH11357. We thank the staff at beamline 19-ID and 23-ID for assistance with data collection. We thank Dr. Steve Lockless for advice on ITC and reading the manuscript, Dr. Matthews Benning for advice on X-ray data processing with Proteum (Bruker), and Tracey Musa for editing the manuscript.

## Author contributions

J.C.S. and Q.S. designed the experiments and wrote the manuscript; Q.S. and X.L. performed the experiments and the data analysis; L.M.P. instructed the ligand docking; and Y.Z. and W.S. contributed to the writing.

## Competing interests

The authors declare no competing interests.
