## [Peer Review File · Nature Communications]

Reviewers' comments:

Reviewer #1 (Remarks to the Author):

Pyrazinamide is a first-line drug that has been used in the treatment of tuberculosis for nearly 70 years. Its role in reducing relapse and treatment rates make it an indispensable component of current TB regimens. Despite its longevity and importance, the mechanism of action has remained elusive until recent years. In this report, the authors follow up on previous indications that the active form of pyrazinamide, pyrazinoic acid, acts by inhibiting L-aspartate decarboxylase (PanD) in the coenzyme A biosynthetic pathway. First, the authors resolve standing issues regarding physical interaction of pyrazinoic acid with PanD using orthogonal approaches. They provide a brief structure activity relationship study with related compounds with differing activity against *M. tuberculosis*. Isothermal titration calorimetry and enzymology are used to confirm interaction and inhibition with complimentary results. Here the authors also present the novel finding that the analog 6-chloropyrazinoic acid has similar activity to pyrazinoic acid. Next, the authors provide compelling structural evidence for binding of pyrazinoic acid in the substrate binding pocket of PanD, which provides a potential explanation for slow tight binding inhibition. Lastly, the authors use structural data to provide a model for the molecular basis of resistance through mutations that have been identified in PanD. Overall, the manuscript presents a rigorous and compelling case for pyrazinoic acid as a slow tight binding inhibitor of *M. tuberculosis* PanD and should enable structure based design of improved inhibitors.

My only concern is that the writing could be improved throughout the manuscript.

Reviewer #2 (Remarks to the Author):

In their manuscript, Sun and colleagues propose the enzyme PanD from *M. tuberculosis* as the real target of the drug pyrazinamide. They present biochemical evidence that pyrazinamide, or rather its product from processing by the enzyme pyrazinamidase, pyrazinoic acid, binds to PanD and inhibits it.

If the findings were true, this would be an important milestone in the drug research against *M. tuberculosis*.

However, the manuscript contains two substantial flaws. The observed inhibition of PanD by pyrazinoic acid is extremely weak, i.e. in the mM range. It remains therefore doubtful, whether such weak inhibition has any physiological effect. I would expect studies on *M. tuberculosis* cells, in order to verify this.

The crystallography seems to be competently done. Nevertheless, the evidence of binding of pyrazinoic acid to PanD is weak, and the electron density shown little convincing. It would be good if the authors would provide Polder-type omit maps in order to demonstrate the binding to the enzyme. Also, since the binding is only mM in nature, it would help to re-do the soaking experiments using higher concentrations of ligands.

Reviewer 1-

1)“...Overall, the manuscript presents a rigorous and compelling case for pyrazinoic acid as a slow tight binding inhibitor of *M. tuberculosis* PanD and should enable structure based design of improved inhibitors.”

Response: We agree.

2) “My only concern is that the writing could be improved throughout the manuscript.”

Response: We have carefully edited the revised manuscript.

Reviewer 2-

1)“If the findings were true, this would be an important milestone in the drug research against *M. tuberculosis*.”

Response: We agree and feel that the explanation below and the new polder omit electron density maps shown at the bottom of the page and in Figure 2 will convince the reviewer that our interpretation of the results are correct.

2)“However, the manuscript contains two substantial flaws. The observed inhibition of PanD by pyrazinoic acid is extremely weak, i.e. in the mM range. It remains therefore doubtful, whether such weak inhibition has any physiological effect. I would expect studies on *M. tuberculosis* cells, in order to verify this.”

Response: Pyrazinamide is known to have poor activity against *M. tuberculosis* (Mtb) in culture, and when it is used as monotherapy in most mouse models of TB infection¹⁻⁵. In fact, in normal culture media, pyrazinamide has no activity against *Mtb*¹. Under growth conditions where the pH is mildly acidic (<pH6) the minimal inhibitory concentration of pyrazinamide is between 0.4-1.6mM (50 – 200 µg/ml)². Importantly, studies have shown that the intracellular pH remains at 7, even in pH 4.5 media^{4,6}. Therefore the 0.8 mM affinity of pyrazinoic acid to PanD is in good agreement with the potency of pyrazinamide/pyrazinoic acid in cell culture. The reason why this first-line drug has low in vitro potency is not known. Some have suggested that pyrazinoic acid accumulates to high levels in the cell, because the acid form cannot be transported out of the cell⁷. Others have speculated that the improved activity at lower pH represents increased penetration of pyrazinamide into the cell⁸. We have included a short explanation about in vivo potency in the introduction of the revised manuscript. Specifically, we have changed the following to the revised manuscript (lines 40-44)...

“Despite the clinical importance of pyrazinamide and the early recognition of its anti-TB activity back in the 1950s, its mechanism of action has not been fully understood^{9,10}. This is primarily because *in vitro* mode of action experiments are complicated by the fact that PZA is not active against *Mtb* cultures grown in normal media¹. It is active under mildly acidic pH, but even then it is not very potent with MICs between 0.4 – 1.6 mM^{2,5}.

3)“The crystallography seems to be competently done. Nevertheless, the evidence of binding of pyrazinoic acid to PanD is weak, and the electron density shown little convincing. It would be good if the authors would provide polder-type omit maps in order to demonstrate the binding to the enzyme. Also, since the binding is only mM in nature, it would help to re-do the soaking experiments using higher concentrations of ligands.”

Response: We greatly appreciate the reviewers' recommendations about using polder omit maps for defining the electron density of the inhibitors. Polder omit maps of the PanD structure with pyrazinoic acid and Cl-pyrazinoic acid, contoured at 3.5 sigma, are of excellent quality (see below and Figure 2) and we hope remove any doubt that we have structures of the complex. The concentration of inhibitors that we soaked/co-crystallized with are between 10-20 mM which is about 18 to 20 times the Kd and therefore we should have >90% bound.

Figure. Polder omit maps for the POA binding sites. a, Close up of the POA binding site. POA is shown with ball-and-stick. The green mesh shows the polder omit map around POA at 3.5σ. Hydrogen bonds around POA are shown in dash-lines. **b, The view is about 60° around the vertical axis. c and d, 6-Cl-POA binding pocket.** The PanD:6-Cl-POA is shown in the same views as the POA complex structure.

1. Tarshis, M.S. & Weed, W.A., Jr. Lack of significant in vitro sensitivity of Mycobacterium tuberculosis to pyrazinamide on three different solid media. *Am Rev Tuberc* **67**, 391-5 (1953).
2. Salfinger, M. & Heifets, L.B. Determination of pyrazinamide MICs for Mycobacterium tuberculosis at different pHs by the radiometric method. *Antimicrob Agents Chemother* **32**, 1002-4 (1988).
3. Lanoix, J.P. et al. Selective Inactivity of Pyrazinamide against Tuberculosis in C3HeB/FeJ Mice Is Best Explained by Neutral pH of Caseum. *Antimicrob Agents Chemother* **60**, 735-43 (2016).
4. Peterson, N.D., Rosen, B.C., Dillon, N.A. & Baughn, A.D. Uncoupling Environmental pH and Intrabacterial Acidification from Pyrazinamide Susceptibility in Mycobacterium tuberculosis. *Antimicrob Agents Chemother* **59**, 7320-6 (2015).
5. Zhang, Y., Permar, S. & Sun, Z. Conditions that may affect the results of susceptibility testing of Mycobacterium tuberculosis to pyrazinamide. *J Med Microbiol* **51**, 42-9 (2002).
6. Vandal, O.H., Pierini, L.M., Schnappinger, D., Nathan, C.F. & Ehrt, S. A membrane protein preserves intrabacterial pH in intraphagosomal Mycobacterium tuberculosis. *Nat Med* **14**, 849-54 (2008).
7. Zhang, Y., Scorpio, A., Nikaido, H. & Sun, Z. Role of acid pH and deficient efflux of pyrazinoic acid in unique susceptibility of Mycobacterium tuberculosis to pyrazinamide. *J Bacteriol* **181**, 2044-9 (1999).
8. Kempker, R.R. et al. Lung Tissue Concentrations of Pyrazinamide among Patients with Drug-Resistant Pulmonary Tuberculosis. *Antimicrob Agents Chemother* **61**(2017).
9. Yeager, R.L., Munroe, W.G. & Dessau, F.I. Pyrazinamide (aldinamide) in the treatment of pulmonary tuberculosis. *Am Rev Tuberc* **65**, 523-46 (1952).
10. Zhang, Y., Shi, W., Zhang, W. & Mitchison, D. Mechanisms of Pyrazinamide Action and Resistance. *Microbiol Spectr* **2**, 1-12 (2013).